# Latent Neural PDE Solver for Time-dependent Systems

**Zijie Li[†], Saurabh Patil[†], Dule Shu, Amir Barati Farimani**
Carnegie Mellon University
Mechanical Engineering Department
{zijieli, ssp2, dules}@andrew.cmu.edu & barati@cmu.edu

## Abstract

Neural networks have shown promising potential in accelerating the numerical simulation of systems governed by partial differential equations (PDEs). Different from many existing neural network surrogates operating on the high-dimensional discretized field, we propose to learn the dynamics of the system in the latent space with much coarser discretization. In our proposed framework, a non-linear autoencoder is first trained to project the full-order representation of the system onto the mesh-reduced space, then a temporal model is trained to predict the future state in this mesh-reduced space. This reduction process simplifies the training of the temporal model by greatly reducing the computational cost with a fine discretization. We study the capability of the proposed framework on 2D/3D fluid flows and showcase that it has competitive performance compared to the model that operates on full-order space.

## 1 Introduction

Many intricate physical processes, from the interaction of protein dynamics to the movement of a celestial body, can be described by time-dependent partial differential equations (PDEs). The simulation of these processes is often conducted by solving these equations numerically, which requires fine discretization to resolve the necessary spatiotemporal domain to reach convergence. Deep neural network surrogates [5, 42, 48, 61, 65, 71] recently emerged as a computationally less-expensive alternative, with the potential to improve the efficiency of simulation by relaxing the requirement for fine discretization and attaining a higher accuracy on coarser grids compared to classical numerical solver [5, 42, 75].

For time-dependent systems, many neural-network-based models address the problem by approximating the solution operator $\mathcal{G}$ that maps the state $u_t$ to $u_{t+\Delta t}$, where the input and output are sampled on discretization grid $\{D_i, D_h\}$ respectively. The input discretization grid can either remain unchanged between every layer inside the network [5, 9, 42], or fit into a hierarchical structure [19, 44, 58, 64, 79, 87] that resembles the V-Cycle in classical multi-grid method. Hierarchical network structures have been a common model architectural choice in the field of image segmentation [70] and generation [25] given their capability for utilizing multi-scale information.

In contrast to the aforementioned approaches especially those that utilize a hierarchical grid structure, our work studies the effect of decoupling dynamics prediction from upsampling/downsampling processes. Specifically, the neural network for predicting the forward dynamics (which we defined as a propagator) only operates on the coarsest resolution, while a deep autoencoder is pre-trained to compress the data from the original discretization grid $D_i$ to the coarse grid $D_l$ (e.g. from a $64 \times 64$ grid to an $8 \times 8$ grid). As the propagator network operates on a lower dimensional space, the training

---

[†]Equal contribution.

cost is greatly reduced and can be potentially adapted to unrolled training with a longer rollout, which is often observed to be helpful to long-term stability [14, 21]. We parameterize the model with a convolutional neural network along with several other components that are popular in neural PDE solvers, including spectral convolution and several variants of attention. We test the proposed framework on 2D and 3D time-dependent PDEs with uniform grids and showcase that the model can achieve efficient data compression and accurate prediction of forward dynamics.

## 2 Related works

**Neural PDE solver**   Neural PDE solvers can be categorized into the following groups based on their model design. The first group employs neural networks with mesh-specific architectures, such as convolutional layers for uniform meshes or graph layers for irregular meshes. These networks learn spatiotemporal correlations within PDE data without the knowledge of the underlying equations [5, 19, 28, 36, 38, 47, 58, 62, 63, 72, 75, 79, 82, 86]. Such a data-driven approach is useful for systems with unknown or partially known physics, such as large-scale climate modeling [33, 53, 59, 66]. The second group, known as Physics-Informed Neural Networks (PINNs) [8, 9, 20, 22, 30, 40, 41, 49, 56, 65, 76, 93], treats neural networks as a representation of the solution function. PINNs incorporate knowledge of governing equations into the loss function, including PDE residuals and consistency with boundary and initial conditions. Unlike the first group, PINNs can be trained solely on equation loss and do not necessarily require input-target data pairs. The third group, known as the neural operators[1, 3, 4, 9, 17, 22, 29, 31, 32, 42, 44, 45, 48, 50, 54], is designed to learn the mapping between function spaces. For a certain family of PDEs, neural operators can generalize and adapt to multiple discretizations without retraining. DeepONet [48] presents a pragmatic implementation of the universal operator approximation theorem[10]. Meanwhile, the concurrent research [43] in the form of the graph neural operator proposes a trainable kernel integral for approximating solution operators in parametric PDEs. Their follow-up work, Fourier Neural Operator (FNO) [42], has demonstrated high accuracy and efficiency in solving specific types of problems. Different function bases such as Fourier[15, 42, 80, 89] / wavelet bases[17], the column vectors from attention layers[9, 40], or Green's function approximation[2, 78], have been be used for operator learning. For more physically consistent predictions[46, 88], neural operator training can be combined with PINN principles.

**Two-stage model for image compression and synthesis**   The utilization of a two-stage model for image synthesis has gained significant attention in the field of computer vision in recent years. VQ-VAEs[67] adopts a two-stage approach for generating images within a latent space. In the initial stage, the approach compresses images into this latent space, using model components such as an encoder, a codebook, and a decoder. Subsequently, in the second stage, a latent model is introduced to predict the latent characteristics of the compressed images, and the decoder from the first stage is used to transform the predicted latent representation back into image pixels. VQ-GANs[13] is developed to scale autoregressive transformers to large image generation by employing adversarial and perceptual objectives for first-stage training. Most recently, several works have developed latent diffusion models with promising results ranging from image[68], point clouds[92] to text generation[35]. Within the domain of neural PDE solvers, the widely employed Encoder-Process-Decoder (EPD) scheme, used to map the input solution at time $t$ to the subsequent time step, stands as a conventional and direct method [6, 27, 57, 61, 71, 74]. As an alternative, researchers have explored propagating the system dynamics in the latent space, aiming to diminish computational complexity and minimize memory usage [34, 90]. Evolving the system dynamics in latent space can involve utilization of recurrent neural networks like LSTM [90], linear propagators grounded in the assumptions of the Koopman operator [37, 51, 52, 55, 77], attention mechanism [24], recurrent MLPs [39] or state-space model [60]. In this work, we propose to use an autoencoder to embed inputs into the latent space, and a simple yet effective convolutional propagator is employed to learn the dynamics of the time-dependent system within this latent space.

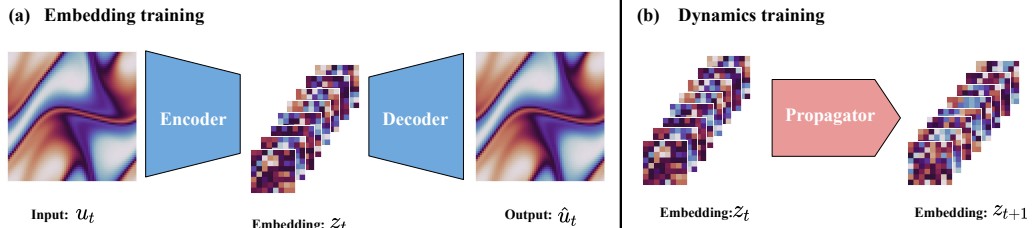

Encoder

Decoder

Input: $u_t$

Embedding: $z_t$

Output: $\hat{u}_t$

(b)  Dynamics training

Propagator

Embedding: $z_t$

Embedding: $z_{t+1}$

Figure 1: (a) An autoencoder is trained to project the input field to latent field with much coarser discretization. (b) A neural network is trained to predict the latent field at different time steps.

# 3 Methodology

## 3.1 Problem definition

We are interested in solving time-dependent PDEs of the following form:

$$\frac{\partial u(\mathbf{x}, t)}{\partial t} = F(u(\mathbf{x}, t), t), \quad \mathbf{x} \in \Omega, t \in [0, T] \tag{1}$$

$$u(\mathbf{x}, 0) = u_0(\mathbf{x}), \quad \mathbf{x} \in \Omega, \tag{2}$$

where $T$ denotes the time horizon and some boundary condition for $\mathbf{x} \in \partial\Omega$ is provided *a priori*. To solve this initial value problem, a neural network is trained to approximate the following mapping:

$$u(\mathbf{x}, t + \Delta t) = \mathcal{A}(u(\mathbf{x}, t)), \tag{3}$$

with a fixed $\Delta t$, and the system is assumed to be Markovian such that $u(\mathbf{x}, t + 2\Delta t) = \mathcal{A}(\mathcal{A}(u(\mathbf{x}, t)))$.

In practice, the function of interest at a particular time step $u(\cdot, t)$ is sampled on a $m$-point discretization grid $D$. For a hierarchical model like U-Net, the grid will be altered internally between different layers and the mapping $\mathcal{A}$ is a composition of a sequence of mapping $\{\mathcal{A}_0, \dots, \mathcal{A}_l\}$ which operates on grids $\{D_0, \dots, D_l\}$ with $D_0 = D$ and the number of grid points $m_l < m_{l-1} < \cdots < m_0$. In contrast to the aforementioned hierarchical model, we propose to learn $\mathcal{A}$ on the coarsest grid $D_l$.

## 3.2 Autoencoder for dimension reduction

One of the most straightforward ways to project the function from the original grid to a coarser grid is interpolation (*e.g.*, bicubic interpolation). However, interpolation can result in significant information loss about the function, as a coarser grid can only evaluate a limited bandwidth and cannot distinguish frequencies that are higher than the Nyquist frequency. To achieve a less lossy compression of the input, we train an encoder network $\phi$ to project the input into latent space when coarsening its spatial grid. In the meantime, we train the decoder network $\psi$ to recover the input from the latent embedding that are represented on the coarse grid. The goal of training these two networks is to achieve data compression without too much loss of information such that their composition approximates an identity mapping: $I \approx \phi \circ \psi$.

In this work, we exploit the fact that the grid structure we are dealing with is uniform and that the majority part of the autoencoder is parameterized with convolutional neural networks (CNN) which are effective for compressing imagery data [13, 69, 84]. On top of the CNNs modules, we also introduced several other modules that have been shown to be effective for PDE surrogate modeling.

**Spectral convolution**    Spectral convolution layer is first proposed in Fourier Neural Operator [42] as a parameterization of the learnable kernel integral [32]. It applies a discrete Fourier transform to the input and then multiplies the $k$-lowest modes with learnable complex weights. Given input function $u_l$, the spectral convolution computes the kernel integral as follows:

$$u_{l+1}(x) = \int_\Omega \kappa(x, y) u_l(y) dy = \sum_{\xi_1=0}^{\xi_1^{\max}} \cdots \sum_{\xi_n=0}^{\xi_n^{\max}} W_j \mathbf{c}_j f_j(x), \quad j = \xi_1 \xi_2 \dots \xi_n \tag{4}$$

where $W \in \mathbb{C}^{(\xi_1^{\max} \times \xi_2^{\max} \times \dots \xi_n^{\max}) \times d_c \times d_c}$ is the learnable weight, $f_j$ is the $j$-th Fourier basis function: $\exp\left(2i\pi \sum_d \frac{x_d \xi_d}{m_d}\right)$ with $m_d$ being the resolution along the $d$-th dimension, $x_d$ being the coordinate for $d$-th dimension, and $\mathbf{c}_j = <u_l, f_j>$ denotes the channel-wise inner product between input function and Fourier series. Unlike the CNN layer, spectral convolution is able to capture multi-scale features that correspond to different frequencies within a single layer. It is also computationally efficient on a uniform grid as the $c_j$ can be efficiently computed via fast Fourier Transformation (FFT). In addition, Gupta and Brandstetter [19] hypothesized that suppressing high-frequency modes with spectral convolution before downsampling can further improve the performance of the network.

**Attention** Scaled-dot product attention [85] has become the state-of-the-art models for natural language processing [7, 11] and computer vision tasks [12] with its capability to capture non-local interactions and compute data-dependent weights. Attention is also closely related to the kernel integral [32] defined in the previous subsection, with its theoretical property on specific PDE problems analyzed in several prior works[9, 16, 31]. Given the $i$-th input feature vector $\mathbf{u}_i$ with channel size $d_c$, the (self-)attention can be defined as:

$$\mathbf{z}_i = \sum_{j=1}^m \alpha_{ij} \mathbf{v}_j, \quad \alpha_{ij} = \frac{\exp\left(\mathbf{q}_i \cdot \mathbf{k}_j / \sqrt{d_c}\right)}{\sum_{s=1}^m \exp\left(\mathbf{q}_i \cdot \mathbf{k}_s / \sqrt{d_c}\right)}, \tag{5}$$

where: $\mathbf{q}_i = W_q \mathbf{u}_i, \mathbf{k}_i = W_k \mathbf{u}_i, \mathbf{v}_i = W_v \mathbf{u}_i$ respectively, and $\{W_q, W_k, W_v\} \in \mathbb{R}^{d_c \times d_c}$ are learnable weights. We plug the self-attention layer into the decoder and investigate its effect on learning the latent embedding.

## 4 Experiments

We test out the proposed model on two time-dependent fluid problems and compared our model to a state-of-the-art neural PDE solver Fourier Neural Operator [42]. For all the problems we sample the data on a spatial grid of resolution 64 along each axis.

### 4.1 Datasets

**2D incompressible flow** The 2D incompressible flow we considered here is the 2D flow dataset proposed in Li et al. [42], which is based on 2D Navier-Stokes equation under vorticity formulation. The voriticity form reads as:

$$\frac{\partial \omega(\mathbf{x}, t)}{\partial t} + \mathbf{u}(\mathbf{x}, t) \cdot \nabla \omega(\mathbf{x}, t) = \nu \nabla^2 \omega(\mathbf{x}, t) + f(\mathbf{x}), \quad \mathbf{x} \in (0, 1)^2, t \in (0, T],$$
$$\nabla \cdot \mathbf{u}(\mathbf{x}, t) = 0, \qquad \mathbf{x} \in (0, 1)^2, t \in [0, T], \tag{6}$$
$$\omega(\mathbf{x}, 0) = \omega_0(\mathbf{x}), \qquad \mathbf{x} \in (0, 1)^2,$$

where $\omega$ denotes vorticity: $\omega := \nabla \times u$, the initial condition $\omega_0$ is sampled from the Gaussian random field, the boundary condition is periodic, the viscosity coefficient $\nu$ is $1e - 4$ and the forcing term is defined as: $f(\mathbf{x}) = 0.1(\sin 2\pi(x_1 + x_2) + \cos 2\pi(x_1 + x_2))$. We are interested in learning to simulate the system (by predicting vorticity) from $t = 5$ to $t = 35$ with 30 seconds of time duration. The reference numerical simulation data is generated via the pseudo-spectral method. The dataset contains 1000 trajectories where we use 900 for training and 100 for testing.

**3D smoke buoyancy** The second benchmark problem is 3D Navier-Stokes equation coupled with advection equation proposed in Li et al. [41] and similar 2D cases have been studied in prior works [3, 18, 81]. The equation describes the motion of rising smoke in a closed box,

$$\frac{\partial \mathbf{u}(\mathbf{x}, t)}{\partial t} + \mathbf{u}(\mathbf{x}, t) \cdot \nabla \mathbf{u}(\mathbf{x}, t) = \nu \nabla^2 \mathbf{u}(\mathbf{x}, t) - \frac{1}{\rho} \nabla p(\mathbf{x}, t) + \mathbf{f}(\mathbf{x}, t), \quad \mathbf{x} \in (0, L)^3, t \in (0, T],$$

$$\frac{\partial d(\mathbf{x}, t)}{\partial t} + \mathbf{u}(\mathbf{x}, t) \cdot \nabla d(\mathbf{x}, t) = 0, \qquad \mathbf{x} \in (0, L)^3, t \in (0, T],$$

$$\nabla \cdot \mathbf{u}(\mathbf{x}, t) = 0, \qquad \mathbf{x} \in (0, L)^3, t \in [0, T],$$

$$\mathbf{u}(\mathbf{x}, 0) = 0, \quad \mathbf{d}(\mathbf{x}, 0) = d_0(\mathbf{x}), \qquad \mathbf{x} \in (0, L)^3,$$

$$\tag{7}$$

where $d$ depicts a marker field for smoke and is subjects to the Neumann boundary condition: $\partial d/\partial n = 0$, the velocity field $\mathbf{u}$ is under Dirichlet boundary condition: $\mathbf{u}(\mathbf{x}, t) = 0, \mathbf{x} \in \partial\Omega$, the initial condition of the marker field $d$ is sampled from a random field, the forcing term is based on the Bousinessq model $\mathbf{f}(\mathbf{x}, t) = [0, 0, \eta d(\mathbf{x}, t)]$ with $\eta$ being the buoyancy factor. We study the case with viscosity coefficient $\nu = 0.003$ and buoyancy factor $\eta = 0.50$. The goal is to predict the marker field and velocity field from $t = 0$ to $t = 12$, with domain size $L = 8$. The reference simulation data is generated using *phiflow* [26] with pressure projection and Maccormack advection scheme [73]. The dataset contains 2200 trajectories amoung which we use 2000 for training and 200 for testing.

## 4.2 Implementation

**Autoencoder** The encoder and decoder are mainly built upon convolutional layers. Internally they comprise a stack of downsampling/upsampling blocks, where each block downsamples/upsamples the spatial resolution by a factor of 2. Each block contains a residual convolution block and a downsampling/upsampling layer. The residual convolution block consists of group normalization [91] and two $3 \times 3$ convolution layers. The downsampling layer uses a $3 \times 3$ convolution layer with a stride of 2, and the upsampling layer upsamples the resolution by using nearest interpolation followed by a $3 \times 3$ convolution layer. We also investigate the influence of inserting spectral convolution layers into each downsampling block and add self-attention layers to the lowest resolution following prior works on image synthesis [13, 68]. For the 2D problem, we set the latent resolution to $8 \times 8$ and the latent dimension to 16. For the 3D problem, we set the latent resolution to $16 \times 16 \times 16$ and the latent dimension to 64. In addition, on the 3D problem, we use the multi-dimensional factorized attention [41] instead of standard attention to reduce the computational cost.

**Propagator** We use a simple residual convolution network [23] to forecast the forward dynamics in the latent space, where each residual block contains a group normalization layer and three convolution layers with $3 \times 3$ convolution kernels. We also employ dilated convolution for the middle convolution layer to capture longer-range interaction. For the 2D problem, we use 3 residual blocks with network width 128. For the 3D problem, we use 4 residual blocks.

**Baseline** On the 2D problem, we tested out two versions of the FNO. The first version is based on the hyperparameter provided in the original paper [42], where the model width is 32 and 8 lowest modes are used at each spectral convolution layer. We also test out a larger version with a width of 64 and use a mode number of 16. On the 3D problem, we use a width of 64 and a mode number of 12 as increasing the mode number for 3D spectral convolution will drastically increase the model parameter (by cubic).

**Training** We first train the autoencoder by minimizing the relative $L^2$ reconstruction loss for around 150k iterations with constant learning rate $3e - 5$ using batch size of $64/16$ respectively for 2D/3D. We then train the propagator by minimizing the mean squared error between predicted embeddings and embedding of reference data for another 150k iteration with a learning rate of $5e - 4$ and a cosine annealing schedule. For FNO we train it with a learning rate $5e - 4$ and a cosine annealing scheduling to minimize the relative $L^2$ prediction loss. The total training iterations are also set to 150k. Different from the original FNO paper, we do not use full rollout during training as we observe reducing the rollout steps during training can significantly improve the performance on NS2D. * We rollout for 2 steps for all models unless stated otherwise.

## 4.3 Results

In this section, we present the comparison between the proposed framework and other models. We observe that the proposed model consistently outperforms FNO which operates on the full mesh space and for lower-dimensional problems like 2D fluid flow the performance gap is more significant. On more complex 3D flow, the model is able to compress the original data to a much coarser (4 times coarser) resolution and learns to predict with accuracy on par with full-order models. Furthermore, as the temporal model operates on a much coarser discretization, we can afford longer rollout training to allow gradient propagated from farther future which can further improve the model's performance on

---

*On 2D Navier-Stokes, FNO (8 modes) has a prediction error of 0.2596 if using fully rollout training, whereas rolling out for 2 steps yields an error of 0.1689.

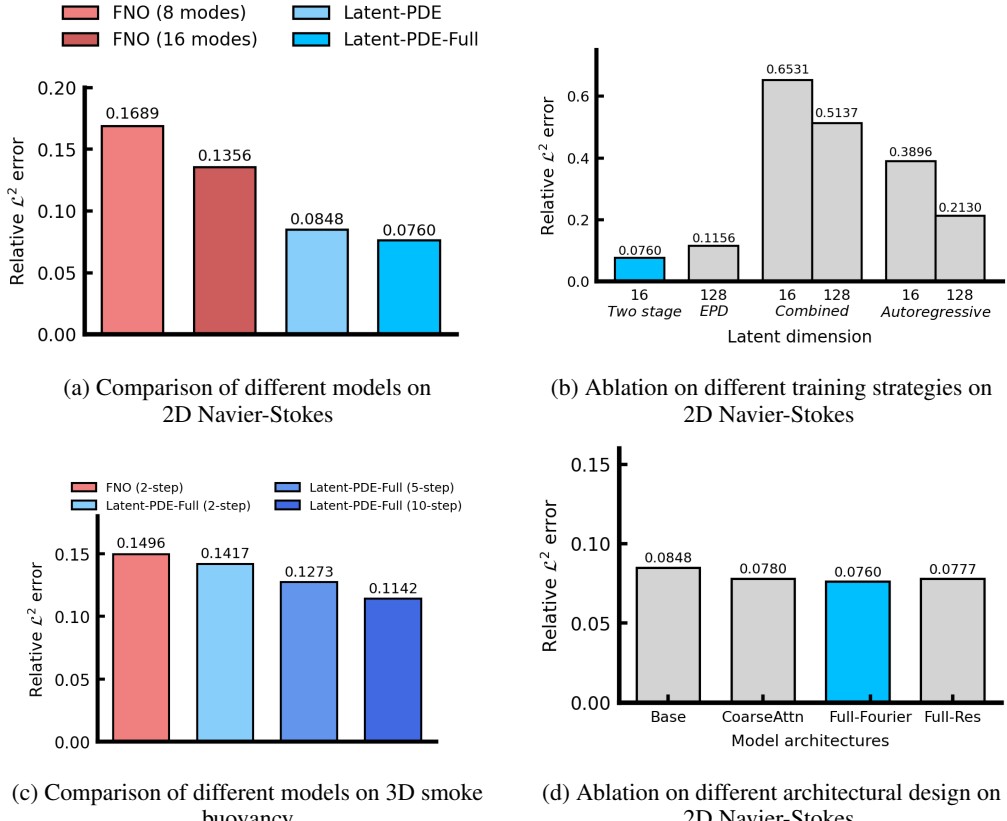

(a) Comparison of different models on 2D Navier-Stokes

(b) Ablation on different training strategies on 2D Navier-Stokes

(c) Comparison of different models on 3D smoke buoyancy

(d) Ablation on different architectural design on 2D Navier-Stokes

Figure 2: Quantitative study on model's performance. Latent-PDE denotes our proposed latent neural PDE solver. "Base" model contains only residual convolutional blocks and fully-connected layers. "CoarseAttn" means we add self-attention to the bottleneck part of the model. "Full-Fourier" means we add spectral convolution layers at the top two downsampling blocks in the encoder and decoder of "CoarseAttn" model. "Res" means we replace spectral convolution layer with residual convolutional blocks. x-step models are rollout for x steps during the training.

predicting the equilibriuim state of the smoke marker field (Figure 2c). (Sample visualization of the best model's prediction are presented in Appendix A)

We also study how different training strategies will influence the model's performance (Figure 2b). We maintain consistent hyperparameters and explore three training strategies: the two-stage method discussed in the previous subsection (referred to as "two-stage"), training the autoencoder and propagator simultaneously by minimizing both reconstruction and prediction loss jointly (referred to as "combined"), and considering the autoencoder and propagator as an unified entity to predict the subsequent step (referred to as "autoregressive"). We also compare two-stage training to Dilated-ResNet [74] that employs a Encode-Process-Decode (EPD) scheme [6, 61, 71] . We find that two-stage training yields the best performance compared to other strategies, which indicates the advantage of two-stage training in obtaining high-quality coarse-graining of the system.

| | FNO2D | Latent-PDE 2D | | FNO3D | Latent-PDE 3D | |
|---|---|---|---|---|---|---|
| | | Autoencoder | Propagator | | Autoencoder | Propagator |
| Fwd + Bwd time (sec) | 0.067 | 0.103 | 0.013 | 2.223 | 1.375 | 0.372 |
| Memory (GB) | 1.87 | 2.54 | 0.25 | 33.33 | 37.15 | 8.10 |
| # of params (M) | 16.8 | 9.7 | 1.4 | 226.5 | 38.8 | 5.4 |

Table 1: Computational cost of different models' training. 2D benchmark is carried out on RTX 3090, using a batch size of 64. 3D benchmark is carried out on A6000, using a batch size of 16.

Compared to FNO that has log-linear complexity with respect to the grid size, the training of the proposed model is relatively slower when combining the time cost for autoencoder training and temporal model training. However, since the temporal model training is much more efficient in latent-pde solver, its training can be less costly on system that requires rolling out for more steps during training.

## 5  Conclusion

In this work, we study a straightforward yet effective data-driven framework for predicting time-dependent PDEs. We show that training the temporal model in the mesh-reduced space improves the computation efficiency and is beneficial for problems that feature latent dynamics distributed on a low-dimensional manifold. The observation in this study is also in alignment with the recent success of a series of image synthesis models that learn the generative model in the latent space instead of pixel space [13, 68, 83]. As this work only considers uniform mesh, an interesting future direction would be the extension to arbitrary meshes and geometries.

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

# A   Sample visualization

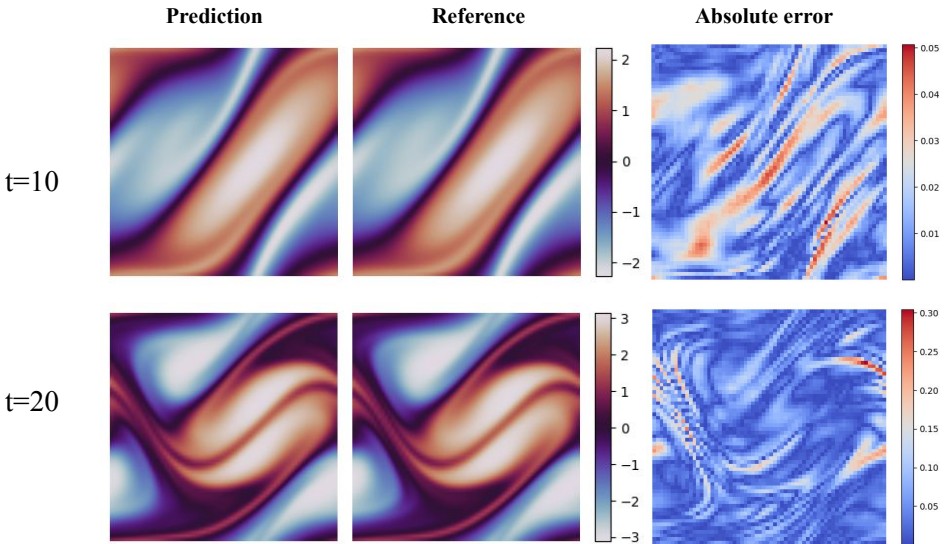

Figure 3: Visualization of model's prediction on 2D Navier-Stokes equation

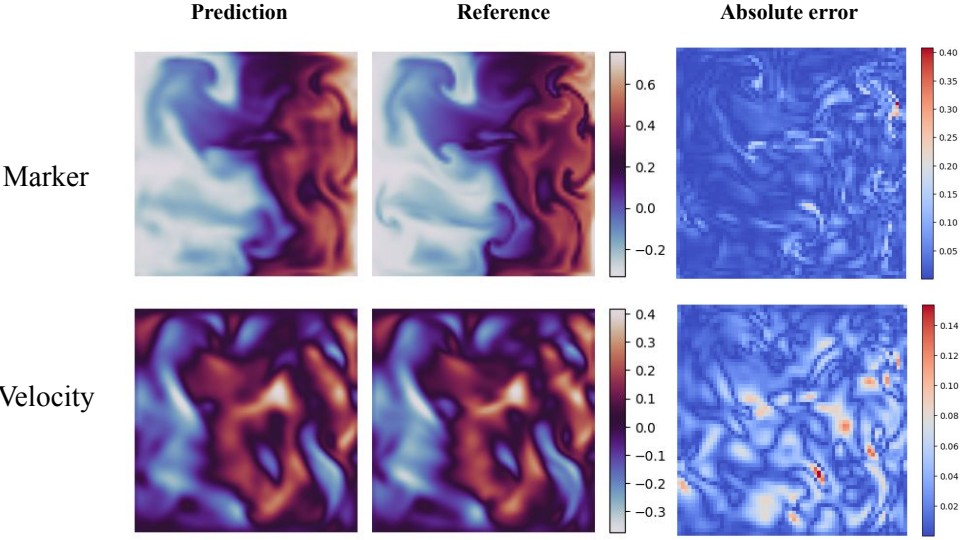

Figure 4: Visualization of model's prediction on 3D smoke buoyancy at cross-section plane $x = 4m$ and time $t = 9s$.

