# OpenReview forum: "Latent Neural PDE Solver for Time-dependent Systems"
_NeurIPS.cc/2023/Workshop/AI4Science — NeurIPS2023-AI4Science Poster_

### Official Review · Reviewer_b2XW · 2023-10-22
**Encoder-Decoder prediction of dynamical systems**

**Rating:** 6
**Confidence:** 4

**Review:**

This paper follows the Encoder-Propagator-Decoder scheme to predict 2d incompressible flow and 3d smoke buoyancy. The encoder-decoder is an autoencoder with self-attention and spectral convolution layers, similar to Latent Diffusion models. The propagator is a ResNet.

Pros:
- Comparisons with Fourier Neural Operator show competitive performace
- Ablations, inclusion of 2d and 3d tasks, comparison of time and memory cost during inference

Things to improve:
- Comparison with any other competitive EPD method from the ones the authors already reference.
- Provide explanations why spectral convolution, self-attention and the proposed architecture in general is a good model choice for the problem (esp. vs FNO and other EPD approaches)
- Training time details are also insufficient, how long does each method take to train?